

# Design of a hazard prediction system with intelligent multimodal fusion based on artificial intelligence & internet of things technology: taking a crib as an example

Jibin Yin[1],[*], Jia'nan Zhao[1],[*] and Xiangliang Zhang[2]

[1] School of Information Engineering and Automation, Kunming University of Science and Technology, Kunming, Yunnan, China
[2] School of Mechanical Engineering, Zhejiang University, Hangzhou, Zhejiang, China
[*] These authors contributed equally to this work.

## ABSTRACT

**Problem:** How to design an intelligent multimodal fusion hazard prediction system using AIoT (Artificial Intelligence & Internet of Things) technology to predict some potential dangers. This article will provide some ideas and methods.

**Introduction:** This article designs an intelligent multimodal fusion hazard prediction system based on AIoT technology. The system mainly consists of an IoT hardware device and an AI multimodal and multi-dimensional Hazard Prediction Algorithm.

**Method:** This article will take baby cribs as an example, using this system to empower traditional baby cribs and transform them into intelligent baby cribs.

**Results:** In this example, the system can detect the real-time status of the baby and predict upcoming dangers, including kicking the quilt, wetting the bed, fever, crying, climbing over the crib, and turning over.

**Hypothesis:** Assuming that intelligent systems can detect the status of infants in real-time and make predictions before potential dangers occur, to promptly alert parents. Perhaps it can prevent some irreversible dangers from occurring.

**Background:** This system solves the pain point problem of parents having to take care of their babies after working hard.

**Purpose:** In this scenario, the system provides a design scheme for an intelligent multimodal fusion hazard prediction system with a temporary care function. The application of this system to baby cribs not only reduces the burden on parents but also ensures the safety and comfort of the baby.

**Transferability:** At the same time, the system has transferability, and its design concept can be transferred to other application scenarios, such as hospital care for infants, young children, or elderly patients, as well as childcare stations for abandoned children in society.

**Significance:** This innovative system design scheme has a positive significance for family harmony and social development.

Corresponding author
Xiangliang Zhang, xlzh@zju.edu.cn

# INTRODUCTION

With the development of Artificial Intelligence & Internet of Things (AIoT), the combination of artificial intelligence and Internet of Things technology has provided new possibilities for young parents to solve the problem of caring for their babies. In modern society, young parents face work pressure and difficulties in infant care (*Honghui et al., 2020*). Leaving the baby alone in the crib may pose potential risks, causing confusion and challenges for parents.

In addition, excessive and inconsolable crying behavior in otherwise healthy infants (a condition called infant colic (IC)) is very distressing to parents, may lead to maternal depression, and in extreme cases, may result in shaken baby syndrome (*Adam-Darque et al., 2020*). At this time, parents need to comfort their children promptly. At the same time, parents also need to try to find the cause of their child's crying, which may be some other factors, such as psychological, physiological, or pathological factors.

Furthermore, there are also some potential risk factors during infant care. For sleeping babies, suffocation is a serious hidden danger, often causing parents to feel worried and anxious. After the baby turns over, there is a risk of suffocation if the face is facing the pillow. Babies may also have a low body temperature due to kicking off the quilt, leading to symptoms of fever, or the risk of falling due to climbing on the edge of the crib. If parents can detect these situations promptly before they occur, perhaps tragedy can be avoided.

However, due to busy daily tasks, parents may find it difficult to constantly monitor their children's health and may miss important indicators such as bedwetting or fever. If a baby doesn't promptly change their diaper after wetting the bed, it may lead to skin infections caused by bacteria or fungi, which can impact the child's immune function. If a baby's fever is not treated promptly, excessive body temperature may lead to dehydration shock (*Farre et al., 2021*). Hyperthermia will lead to a prothrombotic state and, if sufficiently severe such as in heatstroke, a consumption coagulopathy, which will clinically manifest with the simultaneous appearance of intravascular thrombotic obstruction and an increased bleeding tendency (*Levi, 2018*). This brings unnecessary worries and doubts to parents.

To solve the dilemma between young parents taking care of their babies and resting, hazard prediction systems have emerged. The system uses AIOT technology to realize real-time monitoring and data transmission of infant status through multimodal fusion sensors, control units, and Internet connection. For example, sound sensors can monitor a baby's crying, temperature and humidity sensors can monitor the temperature and humidity of the baby's quilt, and Turn-over Detection Algorithms can monitor whether the baby has turned over. By analyzing and processing these data, the system can provide warning sounds to promptly remind parents to handle dangerous situations, such as the risk of suffocation or abnormal body temperature. In addition, the hazard prediction system can also assist in nursing and prevent risks such as crying, bedwetting, and falling of infants.

This article introduces the design of a hazard prediction system based on AIoT technology, using a baby crib as an example for research. By using AIoT technology,

traditional baby cribs can be upgraded to intelligent baby cribs (*Honghui et al., 2020*), providing parents with the convenience of briefly taking care of their babies and ensuring their safety and comfort.

Additionally, the design concept of this system is also transferable and can be applied to other fields, such as providing more convenient care for infants, young children, or elderly patients in hospitals, or improving the efficiency of infant care in childcare stations for abandoned children. The emergence of this design scheme has a positive significance for the progress in the above-mentioned fields.

Outline of this article: this study mainly consists of five parts. The first part is the introduction, which mainly introduces the research background, research objectives, research significance, and the structure of the entire text of the intelligent baby crib. The second part is related work, mainly introducing the research and contributions made by other scholars in the fields of AIoT and baby cribs. The third part is a methodology, which mainly introduces the unique innovation points of this study that are different from existing research, an overview of system principles, module design of IoT devices, and design of hazard prediction algorithm. The fourth part is the experiment, which mainly introduces how the simulation experiment was conducted, including simulating the testing of seven different scenes, recording and exporting data, and analyzing the results of three-dimensional smooth images obtained from the experimental data. The fifth part is a summary, mainly introducing the conclusions drawn from this simulation experiment, subsequent optimization, and improvement, as well as the prospects of this research.

## RELATED WORKS

Extensive research literature in the fields of AIoT and infant physiology has emerged in recent years, providing essential technical support and a basis for hazard prediction systems. Before designing such a system, this article's comprehensive research on AIoT technology, baby physiology, sensors, microcontrollers, and smart homes was conducted to obtain effective references for system design.

*Chang et al.*'s *(2021)* article titled "A Survey of Recent Advances in Edge-Computing-Powered Artificial Intelligence of Things" provides a comprehensive review and in-depth exploration of the core content of AIoT technology. This review explains that the core architecture of AIoT systems typically consists of a perception layer, a network layer, and an application layer, utilizing sensors, wired and wireless networks, and cloud computing to achieve comprehensive perception and connectivity. By applying cloud computing, big data, and artificial intelligence algorithms, AIoT systems can store, analyze, and process massive amounts of data, and achieve intelligent decision-making and control. Additionally, the introduction of edge computing technology enables the system to respond and make decisions more quickly, thus improving real-time performance and efficiency.

*Adli et al.*'s *(2023)* literature review highlights the progress, applications, and advantages of AIoT technology. Their research explores the concept of AIoT, from smart devices to the application of AI technology, providing ideas and directions for this article further research. *Wang et al.*'s *(2023)* research focuses on the application of graph

neural networks in AIoT data collection, proposing a unified GNN pipeline based on the encoder-decoder paradigm and systematically classifying and summarizing emerging technologies. Their research provides new ideas and methods for data collection in the field of AIoT, especially in dealing with noisy and adversarial data, which is highly significant.

Through the review and analysis of the above research results, it is clear that AIoT technology holds vast application prospects in fields such as smart homes, smart factories, and smart agriculture. For instance, smart home systems can intelligently control various devices by sensing the environment and user behavior, providing a more comfortable, safe, and energy-saving living environment. The intelligent factory system can achieve equipment monitoring and predictive maintenance, improving production efficiency and equipment utilization. At the same time, AIoT technology has also shown great potential in the agricultural field, which can improve production efficiency and agricultural product quality, and reduce resource waste, and environmental pollution.

For example, *Chiu et al. (2022)* have developed an AIoT-powered fish feeder that has not only made fish feeding intelligent but also reduced breeding expenses. *Sung, Devi & Hsiao (2022)* have designed a flood observation system to monitor mountain slopes for floods and improve early warning efficiency. *Zhang et al. (2020)* have proposed an AIoT-based real-time monitoring system for tunnel construction that enhances the automation level of the construction process and helps prevent accidents. *Li et al. (2023)* have designed an AIoT-based assistive system to provide convenience for visually impaired individuals, showcasing the potential of AIoT technology in smart living.

There have been several noteworthy findings in the field of infant physiology. Firstly, *Liu, Wu & Zhu (2022)* discovered that newborns are susceptible to hypothermia during the early postpartum period due to various factors such as low-temperature environments and physiological functions. However, *Lo et al. (2022)* found that infants with hypothermia are at risk of contracting severe bacteria and herpes simplex virus. *Levi (2018)* suggests that hypothermia can delay the coagulation process, but its clinical efficacy is not significant as the damage caused by anticoagulation and fibrinolysis seems to be fully balanced. *Ramgopal et al. (2023)* found that the most common definition of hypothermia is body temperature $\leq 36.0\ °C$ and clinical doctors have lower confidence in caring for infants with hypothermia.

Furthermore, *Ipin, Prisma & Nia Maharani*'s *(2020)* study found that bedwetting problems in infants are often caused by factors such as the baby being in a damp state, which may lead to skin irritation and diseases such as eczema. To address this issue, *Franco & Coble (2023)* evaluated a method called GOGOband®, which is a new bedwetting alarm that utilizes real-time heart rate variability (HRV) analysis and artificial intelligence (AI) to create alerts to wake users up before bedwetting. Therefore, it is necessary to detect bedwetting in advance and avoid its impact on the baby promptly.

Additionally, *Ismail & Nallasamy*'s *(2017)* study found that excessive crying is a common phenomenon in young infants, but crying may be the only clinical manifestation of many serious underlying diseases, so careful clinical examination is needed to rule out organic causes. Due to the negative impact of excessive crying on the mental health of

caregivers and infants, *Chua, Setlik & Niklas (2016)* proposed the need to eliminate potential organic causes of "danger signals" and alleviate caregiver anxiety.

Meanwhile, *Matijasic & Plesa Premilovac*'s *(2018)* study found that crying is a part of the physiological and neural development of infants, but it may also cause anxiety, fatigue, and concern among caregivers. Therefore, predicting a baby's crying in advance and notifying parents promptly to comfort the child may help avoid excessive crying and related diseases.

Moreover, falls are a common cause of head injuries in infants. *Khalid et al. (2019)* pointed out that head injuries in children are the most common cause of death or permanent disability. To prevent infants from falling off the bed and getting injured, *Ganga et al. (2023)* mentioned the minimum rail height requirement set by the Consumer Safety Products Committee (CPSC).

However, in the United States, head injuries related to crib falls are still common. Therefore, issuing warnings to parents before predicting a tendency for babies to climb over cribs may help prevent tragedies from occurring. In addition, *Gaw et al. (2017)* found that approximately 67.3% of infants in early death cases are found in cribs or cradles, and pillows are one of the most related objects to infant asphyxia.

Because accidental suffocation can be largely prevented, both the public and individuals have made tremendous efforts, especially through safer sleep environments, to avoid this situation. Therefore, if it is possible to predict in advance that the baby's face is covered by a pillow and notify parents promptly to deal with this dangerous situation, it may avoid suffocation and prevent the occurrence of tragedy. Finally, hyperthermia can increase the risk of infant dehydration. *Chiu et al. (2023)* found that premature infants with high body temperature upon admission are more likely to develop necrotizing enterocolitis and neurodevelopmental disorders. *Levi (2018)* found that high body temperature can lead to thrombosis and other serious infections. Therefore, if the baby's temperature can be monitored in real-time, parents can be warned when the temperature rises, and this dangerous situation can be dealt with promptly, potentially avoiding major diseases or even death.

Based on the above-mentioned studies, this article designs a hazard prediction system with intelligent multimodal fusion based on AIoT technology to address the issue of short-term care for infants. The system can monitor the baby's status in real-time, such as wetting the bed, kicking off the quilt, fever, crying, climbing over the crib, turning over, and normal status, and alerting parents in dangerous situations.

Compared with the contributions made by other researchers, the difference of this study is that it integrates AIoT technology with multimodal data to provide real-time monitoring and active warning, solving the problem of short-term care, filling the gaps in existing systems, and providing a feasible method and idea for the design of more advanced hazard prediction systems in the future.

## METHODOLOGY

The intelligent multimodal fusion hazard prediction system based on AIoT technology designed in this article is shown in Fig. 1. The system mainly consists of an IoT hardware

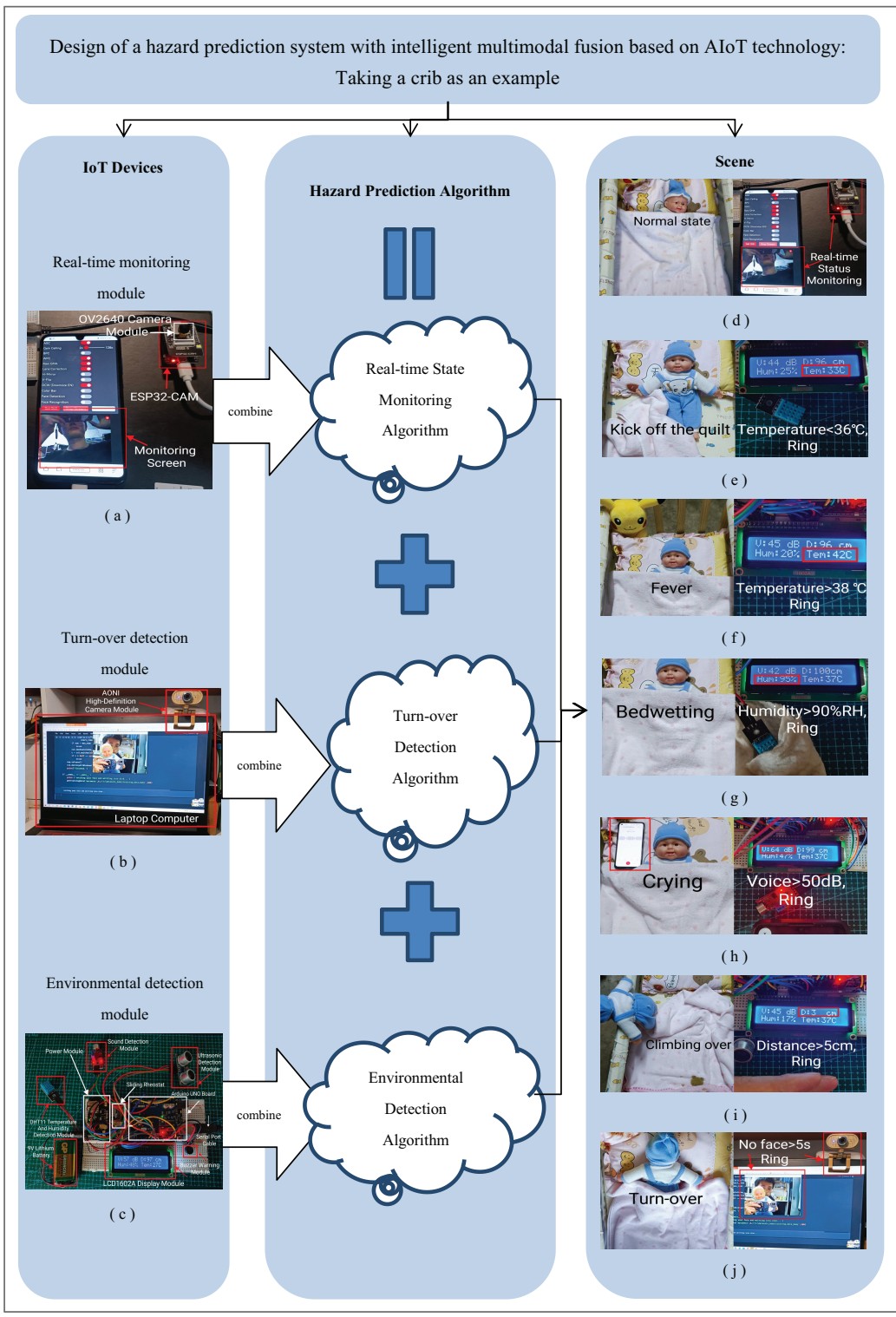

**Figure 1  A hazard prediction system with intelligent multimodal fusion based on AIoT technology.**

device and an AI multimodal and multi-dimensional fusion Hazard Prediction Algorithm. This system can be applied to seven different scenes.

## Main innovation points

(1) The system solves the pain point problem of parents having to take care of their baby's safety after tiring work;

(2) The system utilizes AIoT and multimodal fusion technology to empower traditional baby cribs, enabling ordinary baby cribs to become intelligent baby cribs;

(3) The system has transferability and can transfer the design concept of this hazard prediction system to other application scenarios, such as the care of infants, young children, or elderly patients in hospitals, as well as childcare stations for abandoned children in society.

## System principles

The principle of this system is: firstly, by using the Real-time State Monitoring Algorithm combined with ESP32-CAM and OV2640 cameras, the real-time activity status of infants can be observed through mobile devices. Secondly, by using the Turn-over Detection Algorithm and AONI high-definition camera ('AONI' is a specific product brand.) to run on the PC, it can determine whether the baby has turned over and avoid suffocation. Then, using the Environmental Detection Algorithm with Arduino UNO and various sensors to detect indicators such as temperature, humidity, sound, and distance. Finally, by integrating the above three algorithms, a multimodal Hazard Prediction Algorithm was obtained that can comprehensively analyze the state of infants from multiple dimensions. Through this comprehensive algorithm, not only can the real-time status of the baby be detected, but it can also predict upcoming dangers, including kicking the quilt, wetting the bed, fever, crying, climbing over the crib, and turning over. It can also trigger a buzzer to sound before the danger occurs, reminding parents to handle it promptly. At the end of this experiment, the practicality and effectiveness of the system will be verified through simulation experiments, providing valuable references for subsequent research on hazard prediction systems.

## Module design of IoT devices

### Design of real-time monitoring module

This research has employed the ESP32-CAM development board to achieve real-time monitoring of infants' activity status. The board also transmits video stream information to parents' mobile phones. The ESP32-CAM development board incorporates WiFi and Bluetooth modules, which makes it easy to connect to the internet. The board's camera can provide panoramic real-time monitoring of the baby's condition and transmit the monitoring footage to parents' mobile phones. The key components and functions of the ESP32-CAM development board module circuit design include:

Firstly, a 5V DC power supply is used, and the voltage is stabilized at 3.3V through an AMS117-3.3V voltage regulator chip. This provides the required power for ESP32-CAM and the camera.

Secondly, the ESP32-CAM module is the core aspect of the whole development board. It includes the ESP32 chip (*Gaikwad, 2023*), peripheral circuits, and WiFi module. The ESP32 chip supports WiFi and Bluetooth communication and offers multiple GPIO interfaces that can be connected to other peripherals.

Additionally, it is equipped with a OV2640 camera module (*Gaikwad, 2023*), which supports static image shooting and video recording with a maximum resolution of 1,600 × 1,200 pixels. This camera can efficiently and accurately monitor the baby's activity status and transmit real-time images to the mobile phone. For the convenience of developer communication and debugging, a USB to serial port chip is also configured, allowing the developer to interact with the ESP32-CAM module through the USB interface.

At the same time, it supports functions such as microSD card expansion (*Gaikwad, 2023*) and GPIO expansion and can be expanded and connected as needed. Through the baby activity monitoring system designed in this way, parents can keep up-to-date with the baby's situation through a stable internet connection and provide timely attention and care. The design of the real-time monitoring module is shown in Fig. 1A.

### Design of turn-over detection module

Turn-over Detection Module design consists of multiple components including the AONI high-definition camera, interfaces, and the PC terminal. These components collectively form the Turn-over Detection Module of the system, which is capable of precise facial detection of infants by leveraging the AONI camera's interface connected to the PC terminal. The design of the turn-over detection module is shown in Fig. 1B.

The PC terminal serves as the central processing unit of the system, orchestrating the various stages of the turn-over detection workflow. Equipped with robust computational capabilities, the terminal operates in tandem with the AONI camera, receiving real-time image data and invoking the turn-over detection algorithm. Leveraging sophisticated image processing techniques, the algorithm employs facial detection algorithms to accurately detection and track the infant's facial features.

The integration of this module with the Turn-over Detection Algorithm enables accurate facial detection of infants. By examining the presence or absence of detected facial images, it becomes possible to determine if the infant has turned over. If no facial image is detected within a 5-s interval, it indicates that the infant has turned over and their mouth and nose might be obstructed. In response, the system automatically triggers an alert through a buzzer, prompting parents to take immediate action.

The synergy between the AONI camera, the PC terminal, and the turn-over detection algorithm ensures a robust and reliable Turn-over Detection Module for infant safety. This comprehensive approach not only guarantees real-time monitoring of infants but also provides timely alerts to parents, enabling them to promptly address potential risks associated with the infant's position during sleep or rest.

Future research endeavors may focus on further enhancing the performance of the Turn-over Detection Module. This could include exploring novel image processing techniques, refining deep learning models, and integrating additional sensor modalities for a more comprehensive understanding of the infant's behavior and environment.

**Table 1 The equipment components of the entire hazard prediction system and their roles.**

| No. | Name of component | The task of the components in the device |
|---|---|---|
| 1 | Arduino UNO board | The main function of the Arduino UNO development board module is to control the power module, LCD1602A display module, temperature and humidity detection module, sound detection module, ultrasonic detection module, and buzzer warning module to work together, responsible for collecting temperature, humidity, decibel value, distance value, and controlling the opening or closing of the buzzer. |
| 2 | 9V lithium battery and power module | The power module can provide stable power supply for the entire system and can convert $9V$ power into $5V$ or $3.3V$ for output power supply. |
| 3 | LCD1602A display module | It can display relevant information about the baby, such as temperature, humidity, decibel values, and distance values. |
| 4 | DHT11 temperature and humidity detection module | Measure temperature and humidity. |
| 5 | KY-037 sound detection module | Measure decibel value. |
| 6 | HC-SRO4 ultrasonic detection module | Measure distance value. |
| 7 | Buzzer warning module | Make a sound to provide a warning signal. |
| 8 | ESP32-CAM development board module | Responsible for recording real-time video streaming information of the baby, and connecting to the phone through WiFi hotspots to transmit the video back to the parents' phone. |
| 9 | OV2640 camera module | Record video stream. |
| 10 | AONI high-definition camera module | Combining AI facial recognition algorithms to detect whether a baby has turned over. |
| 11 | Sliding Rheostat | By changing its own resistance value, the current and voltage flowing through the entire system can be altered. |

## Design of environmental detection module

The design of the environmental detection module is shown in Fig. 1C. It consists of various components, including the main control chip Arduino UNO development board module, power module, LCD1602A display module, temperature and humidity detection module, sound detection module, ultrasonic detection module, buzzer warning module, ESP32-CAM development board module, OV2640 camera module, AONI high-definition camera module, and sliding resistor. The functions of these modules in the entire hazard prediction system are listed in above Table 1.

The detailed explanation of the Table 1 is as follows: The main function of the Arduino UNO development board module is to control the power module, LCD1602A display module, temperature and humidity detection module, sound detection module, ultrasonic detection module, and buzzer alarm module to work together, responsible for collecting temperature, humidity, decibel value, distance value, and controlling the opening or closing of the buzzer. The power module can provide a stable power supply for the entire system and can convert 9V power into 5V or 3.3V for output power supply.

The DHT11 temperature and humidity detection module, a sound detection module, and an ultrasonic detection module can monitor the relevant information of infants in real-time, such as temperature, humidity, decibel value, and distance value, while the LCD1602A display module can dynamically display and refresh these four data in real-time. The buzzer warning module can issue an alarm when needed to remind parents or

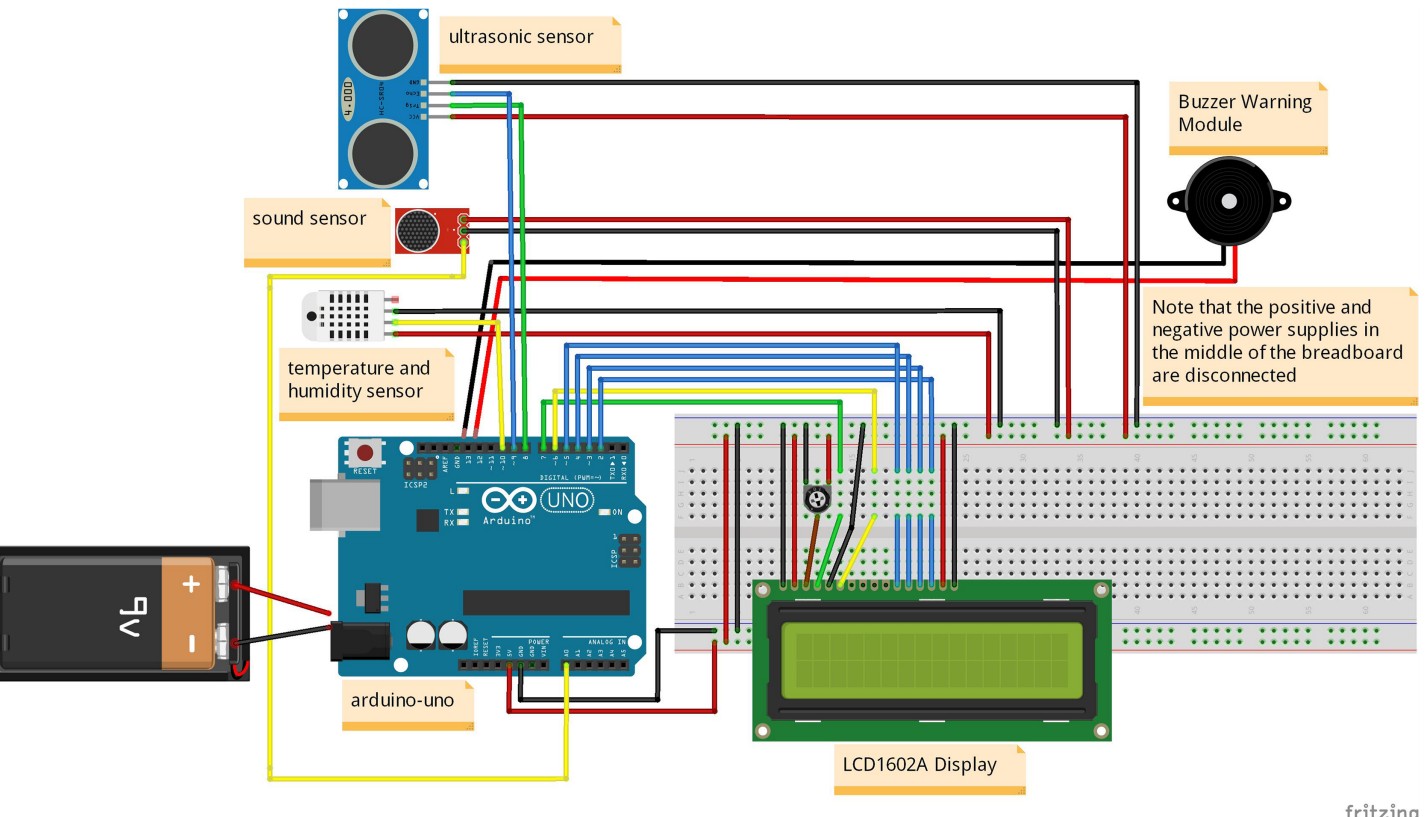

**Figure 2 Simulation design diagram of environmental detection module.**

guardians. Finally, the environment detection module was simulated and designed using the simulation software Fritzing, as shown in Fig. 2.

This simulation design diagram provides a detailed explanation of the composition structure of the environmental detection module, the wiring relationships of various sensors, and the corresponding pin ports, which is conducive to landing the physical object and conducting scene simulation experiments.

## Design of hazard prediction algorithm
### Design of real-time state monitoring algorithm

Firstly, initialize and configure the camera. If the camera model is "ESP32-CAM", set pins 13 and 14 to input pull-up. Then, initialize the camera and check if the initialization is successful. If initialization fails, please print an error message and end the algorithm. Next, initialize the sensor and make some settings based on the sensor model. Then, check if the LED pins are defined. If defined, please set LED flashing. Then, disable sleep mode and try to connect to WiFi until the connection is successful. Next, start the camera server to transmit the video stream in real-time.

In the main loop, the algorithm will loop infinitely, with each loop delayed by 10 s. This can continuously monitor real-time status and process or display the required video

stream data. This algorithm implements a complete real-time status monitoring system that can initialize cameras, sensors, and LEDs, connect to WiFi, and start the camera server to provide real-time video streams and local IP addresses. Finally, print the HTTP address so that other devices can connect to the camera through it. Finally, opening the IP address through the browser (*Gaikwad, 2023*) allows for real-time viewing of the video screen. The Real-time State Monitoring Algorithm will be presented and analyzed in the form of pseudocode is shown in Algorithm 1.

Important reminder: Before using the ESP32-CAM development board, please make sure to open the WiFi hotspot in advance, otherwise an error message of "WiFi connection failure" may appear. Next, burn the program onto the ESP32-CAM development board and press the RST button on the board to restart it. At this point, the core program will automatically run and connect to a WiFi hotspot LAN named "XXX" with a password of "XXX". Next, select the serial port monitor under the Arduino IDE compiler interface tool to view the IP address output from the serial port.

In addition, the above program is only applicable to local area networks. If you want to remotely monitor real-time videos within a wide area network, you can try the following approach: design two subroutines, one for running on the ESP32-CAM development board and the other for running on the server. Send video data to the server through the ESP32-CAM development board, and the server runs the receiving program to receive and display it.

The advantage of real-time video stream information being sent to external public network servers is that distant parents can receive remote real-time video monitoring on their mobile phones.

However, it should be noted that even if the video of the impending danger is transmitted back to parents, they will not be able to rush home in time to prevent the tragedy from happening. Therefore, this study only focuses on the scenario where parents are at home (within a local area network), aiming to provide the effect of short-term care.

### Design of turn-over detection algorithm

The implementation logic of the Turn over Detection Algorithm is to first create a window to display real-time images captured by the camera. Then load a human face classifier model, which is used to detect faces in the image. In each frame of the image, the program converts the image into a grayscale image and uses a classifier model to detect faces.

If a face is detected, the program will record the current time and add the data to the list of duration and status. The detected face will be marked with a green box on the image. If no face is detected, the program will also record the current time and add the data to the list of duration and status, while setting the status to "Turn over" to indicate flipping. If no face is detected for more than 5 s and the buzzer has not been triggered, trigger the buzzer and set its triggering status to triggered.

After the loop ends, the program will write the list data of duration and status into an Excel spreadsheet. It uses the openpyxl library to create a new workbook and creates column headings on the active table.

**Algorithm 1    Real-time state monitoring algorithm.**

**Input:** camera_model, ESP32-CAM_pin, LED_pin, WiFi_SSID, password

**Output:** LocalIP, Real-time state video streaming

1. **procedure** INITIALIZECAMERA
2.          **if** camera_model is "ESP32_CAM" **then**
3.               set pin 13 and pin 14 as input pull-up
4.          **end if**
5.          Initialize the camera and apply the configuration
6.          **if** camera initialization fails **then**
7.               Print the error message and return
8.          **end if**
9. end procedure
10. **procedure** INITIALIZESENSOR
11.          Get the sensor instance
12.          Perform some initialization settings based on the sensor model
13. **end procedure**
14. **procedure** SETUPLED
15.          **if** LED_pin is defined **then**
16.               Set up the LED flash
17.          **end if**
18. **end procedure**
19. **procedure** CONNECTTOWIFI
20.          Disable sleep mode
21.          Connect to WiFi and keep trying until successful
22. **end procedure**
23. **procedure** STARTCAMERASERVER
24.          Start the camera server
25. **end procedure**
26. **procedure** PRINTLOCALIP
27.          Print the local IP address for connecting to the camera
28. **end procedure**
29. INITIALIZECAMERA
30. INITIALIZESENSOR
31. SetupLED
32. CONNECTTOWIFI
33. STARTCAMERASERVER
34. PRINTLOCALIP
35. **while** True **do**
36.          Enter the main loop with a 10-s delay
37. **end while**

| Algorithm 2 | Turn-over detection algorithm. |
|---|---|

1. **FUNCTION** detect_faces(window_name, camera_id, excel_path)
2.     **INITIALIZE** video capture and face classifier
3.     **CREATE Excel** workbook and set headers
4.     **WHILE** capturing video **AND** data_count < 10,000 DO
5.       **READ** frame from camera
6.       **IF** frame is valid **THEN**
7.         **CONVERT** frame to grayscale
8.         **DETECT** faces in the frame
9.         **IF** faces detected **THEN**
10.           **RECORD** current time and update last_face_time
11.           **SET** status to 'Normal'
12.           **DRAW** rectangles around detected faces
13.         **ELSE**
14.           **RECORD** current time
15.           **SET** status to 'Turn over'
16.           **IF** time since last face > 5 s THEN
17.             TRIGGER buzzer
18.           **ENDIF**
19.         **ENDIF**
20.         **DISPLAY** frame
21.         **WAIT** for user input
22.       **INCREMENT** data_count
23.     **SAVE** data to Excel file
24.     **PRINT** 'Finished.'
25. **MAIN FUNCTION**
26.     **CALL** detect_faces with parameters 'mycamera', 0, and Excel file path

Then, it traverses the data in the list and writes the data into the corresponding cells of the table. At the same time, based on the value of the status, set the "Buzzer" column to "Ring" or "Non". Finally, the program saves the Excel spreadsheet and prints "Finished." to indicate that the program has completed execution.

This program implements functions such as capturing real-time images from the camera, facial detection, data recording, and writing Excel spreadsheets. It can be used to track the duration of facial appearance, record facial status, and trigger a buzzer to remind you to turn over. Finally, the data written into an Excel spreadsheet will be used for subsequent experiments and analysis on the accuracy of the Turn over Detection function. The pseudocode logic diagram of Turn-over Detection Algorithm is shown in Algorithm 2.

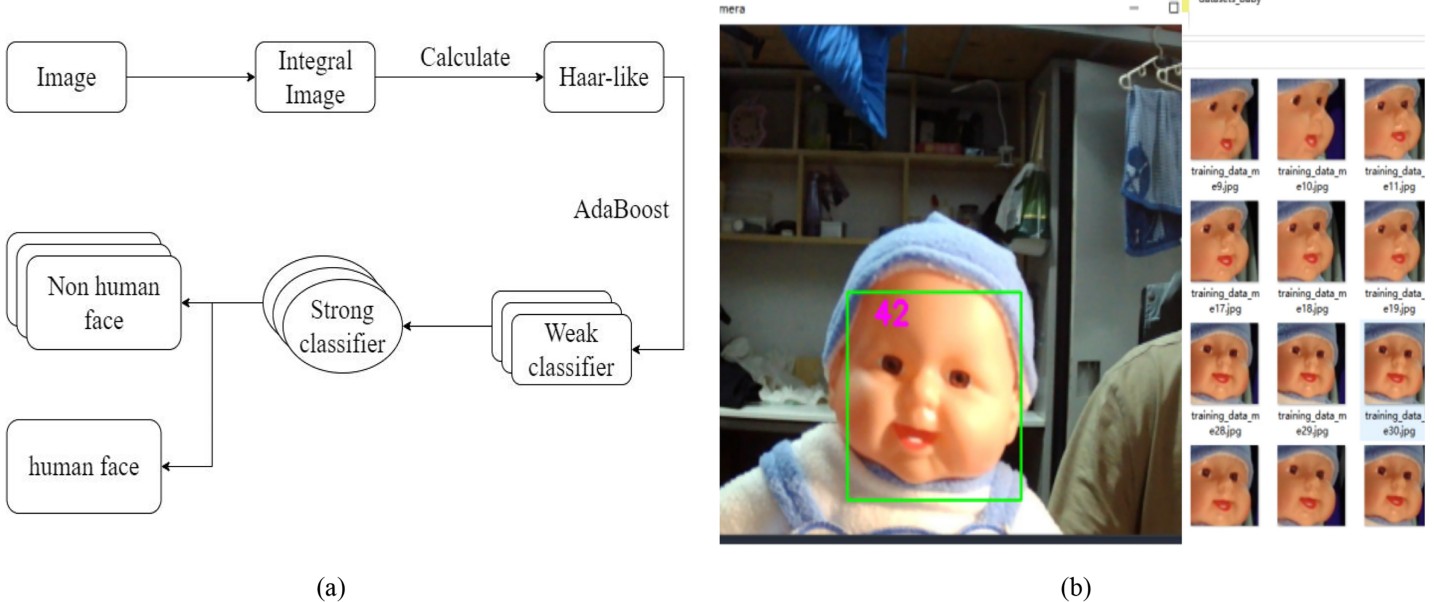

(a) (b)

**Figure 3 Design of turn-over detection algorithm.** Among them, (A) is the workflow diagram of the face detection classifier, and (B) is a demonstration of the effectiveness of the baby turn-over detection algorithm.

This study uses Turn over Detection Algorithm to achieve bounding box detection of infants' faces and determine whether the infant has turned over. In the hazard prediction system, this system chooses to call the face detection algorithm in the Open CV library for face detection. There have been many studies on face detection. Open CV is an open-source library for visualization algorithms and image processing developed by Intel. It has become quite mature and has been widely applied in face detection and recognition (*Lin et al., 2021*). This article selects the Haar cascade classifier. The file of the classifier contains Haar-like features that describe different parts of the human body. It achieves the classification of faces and nonfaces through integrated images, AdaBoost algorithm, and cascaded classifiers (*Lin et al., 2021*).

The main process of the Haar classifier is to use sliding windows and integral images to achieve fast traversal of grayscale images and calculation of Haar-like features. Next, the AdaBoost algorithm based on Haar-like features is used to train a weak facial classifier and construct a strong classifier. Then, multiple strong classifiers are combined to enhance the classification performance (*Lin et al., 2021*). Therefore, facial classification can be achieved, and the workflow of the facial classifier is shown in Fig. 3A.

Object detection using Haar cascaded classifier features is a useful method for object detection. This is a machine learning-based method, where the cascade function is trained from multiple positive and negative images. Each section is a single value obtained by subtracting the number of pixels below the white box from the number of pixels below the black box.

The program will automatically search to determine Haar features based on the lines and edges of dark and bright images (*Diyasa et al., 2021*). There are several types of Haar

features, namely edge features, line features, and center features. After determining the Haar features in a specific image region, the image integration will be calculated within that pixel region (*Diyasa et al., 2021*). The basic calculation process for the image of the Haar cascade classifier is shown in the following Eq. (1) (*Diyasa et al., 2021*):

$$s(x, y) = i(x, y) + s(x, y) + s(x - 1, y) - s(x - 1, y - 1) \tag{1}$$

among them, $s(x, y)$ represents the sum of each pixel value, $i(x, y)$ represents the pixel value intensity value from the input image, $s(x - 1, y)$ represents the pixel value on the $x$-axis, $s(x, y - 1)$ represents the pixel value on the $y$-axis, and $s(x - 1, y - 1)$ represents the pixel value on the diagonal (*Diyasa et al., 2021*). The facial detection effect after running the baby turn-over detection algorithm is shown in Fig. 3B.

An image, contains many Haar features, so a large amount of image integration calculations will also be performed. Afterward, the images will be processed through a cascaded classifier. Cascade classifiers process many features through hierarchical classification (*Diyasa et al., 2021*).

During the face detection process, the system will automatically search for facial information. If there is a face in the field of vision, the system can detect it within 0.04 s. By using multi-scale algorithms to search facial images at low resolution, the system switches to high-resolution search only when a face-like shape is detected.

In addition, future research may consider replacing facial detection algorithms with FaceIt (*Brey, 2004*) recognition systems, which use local feature analysis (LFA) algorithms (*Kim et al., 2020*) to encode facial features and generate facial patterns as unique numerical codes. The system can compare the detected facial patterns with the facial pattern data stored in the database to achieve accurate facial recognition.

### Design of environmental detection algorithm

The environment detection algorithm designed in this article utilizes multiple sensors to work together in multimodal mode and monitor the temperature, humidity, sound intensity, and distance of the environment in real-time. Firstly, the values read by each sensor are printed out on the serial port and converted into strings for real-time display on the LCD screen. Through LCD and serial port output, users can conveniently observe the temperature and humidity, sound intensity, and distance information of the environment.

In addition, the algorithm also provides feedback and control to the system based on certain judgment conditions. When the sound intensity exceeds 50, the temperature exceeds 38 degrees Celsius or is below 36 degrees Celsius, the humidity exceeds 90%, or the distance is less than 5 centimeters, the system will control the buzzer to sound, indicating that danger is about to occur, and output a "Ring" on the serial port. On the contrary, it indicates that the environment is normal and "Non" is output on the serial port. These valid data will be imported from the serial port into an Excel spreadsheet for verification and analysis of subsequent experimental results. The pseudocode logic diagram of Environment Detection Algorithm is shown in Algorithm 3.

**Algorithm 3** Environmental detection algorithm.

**Input:** lcdPin, dhtPin, ledPin, threshold

**Output:** SoundValue, temperature, humidity, distance, Ring/Non

1. **while** *true* **do**

 // Initialization InitializeLCD(*lcdPin, dhtPin*)

2. TurnOnBacklight()

3. InitializeSerialCommunication(*9,600*)

4. InitializeLED(*ledPin*)

5. InitializeSensors()

 // Main Loop **while** *true* **do**

  // Read sound sensor value ← ReadSoundValue()

6. currentTime ← GetCurrentTime()

7. seconds ← currentTime/1,000

8. PrintRuntimeAndSoundValue(*seconds, value*)

9. Delay(*50*)

  // Read distance sensor SendUltrasonicSignal()

10. distance ← GetDistance()

11. Distance: %d *cm*

  // Read temperature and humidity chk ← ReadDHTData()

12. CheckSensorValue(*chk*)

13. **if** *chk = OK* **then**

14.  PrintTemperatureAndHumidity()

15. **end if**

16. Delay(*200*)

  // Display data on LCD temperatureStr, humidityStr ← ConvertToStrings()

17. DisplayOnLCD(*temperatureStr, humidityStr*)

18. Delay(*1000*)

19. ClearLCD()

  // Check conditions and turn on/off the LED

20. **if** CheckConditions(*SoundValue, temperature, humidity, distance, threshold*) **then**

21.  TurnOnLED()

22.  Ring

23.  Delay(*200*)

24. **end if**

25. **else**

26.  Non

27. **end if**

28. **end while**

29. **end while**

# EXPERIMENT

This chapter mainly introduces how simulation experiments are conducted and the experimental results. The next experiment will simulate seven different scenarios. This includes the real-time status of the baby, baby hypothermia, baby hyperthermia, baby bedwetting, baby crying, baby climbing over the crib, and baby turning over. When parents hear a buzzer warning, they need to deal with these special situations promptly. The scene simulation is shown in Fig. 1. The three-dimensional data analysis diagram of the experimental results is shown in Fig. 4. Please refer to Fig. 5 for the red color blindness and Fig. 6 for the green color blindness.

## Functional testing and data analysis

In the design and experimental process, it is necessary to consider the influence of various factors on the experimental results. To ensure the accuracy and reliability of the experiment, the control variable method was adopted. For example, testing only temperature changes while maintaining constant humidity to demonstrate the feasibility of this solution. Variable control and detailed analysis were conducted on various factors such as temperature, humidity, sound intensity, and distance values. Control and analysis have avoided the cross-influence of these factors on the experimental results, ensuring the reliability and accuracy of the experiment. This study conducted a series of simulation experiments, collecting data from various sensors in the system and importing Excel spreadsheets through the Arduino compiler serial port. Subsequently, the data is preprocessed, including column splitting, removing excess space characters, removing letters and symbols, adding column headings, and calculating the data. Finally, the professional tool Origin was used to visualize and analyze the data, draw conclusions, and verify the feasibility and practicality of the system. Next, simulation experiments will be conducted on each of the seven functions of this hazard prediction system.

### Test the function of monitoring the real-time status of infant

During this test, aimed to assess its capability to monitor the real-time status of infants and transmit the monitoring image to parents' mobile phones *via* the Internet.

To simulate different activity states of the baby, the Real-time Monitoring Module was utilized to capture the real-time panoramic view of the infant. After running the Real-time Monitoring Algorithm, the website address will be output. Open the IP address (*Gaikwad, 2023*) through a browser to view the baby's real-time activity scene. The test results indicated that all the baby's indicators remained within the normal range, and the video transmission was successful. Consequently, the functional test for simulating and monitoring the real-time status of infants is deemed successful, as depicted in Fig. 1D.

Table 2 shows real-time recorded values of various indicators for a small subset of randomly selected infants. The distance value in the third and fourth rows of the table above is 0, which is an abnormal situation. This situation may be due to the ultrasonic sensor being obstructed, the distance value exceeding the range, or the refresh delay causing the value to be 0 during the operation process. Among the 1,107 actual measured

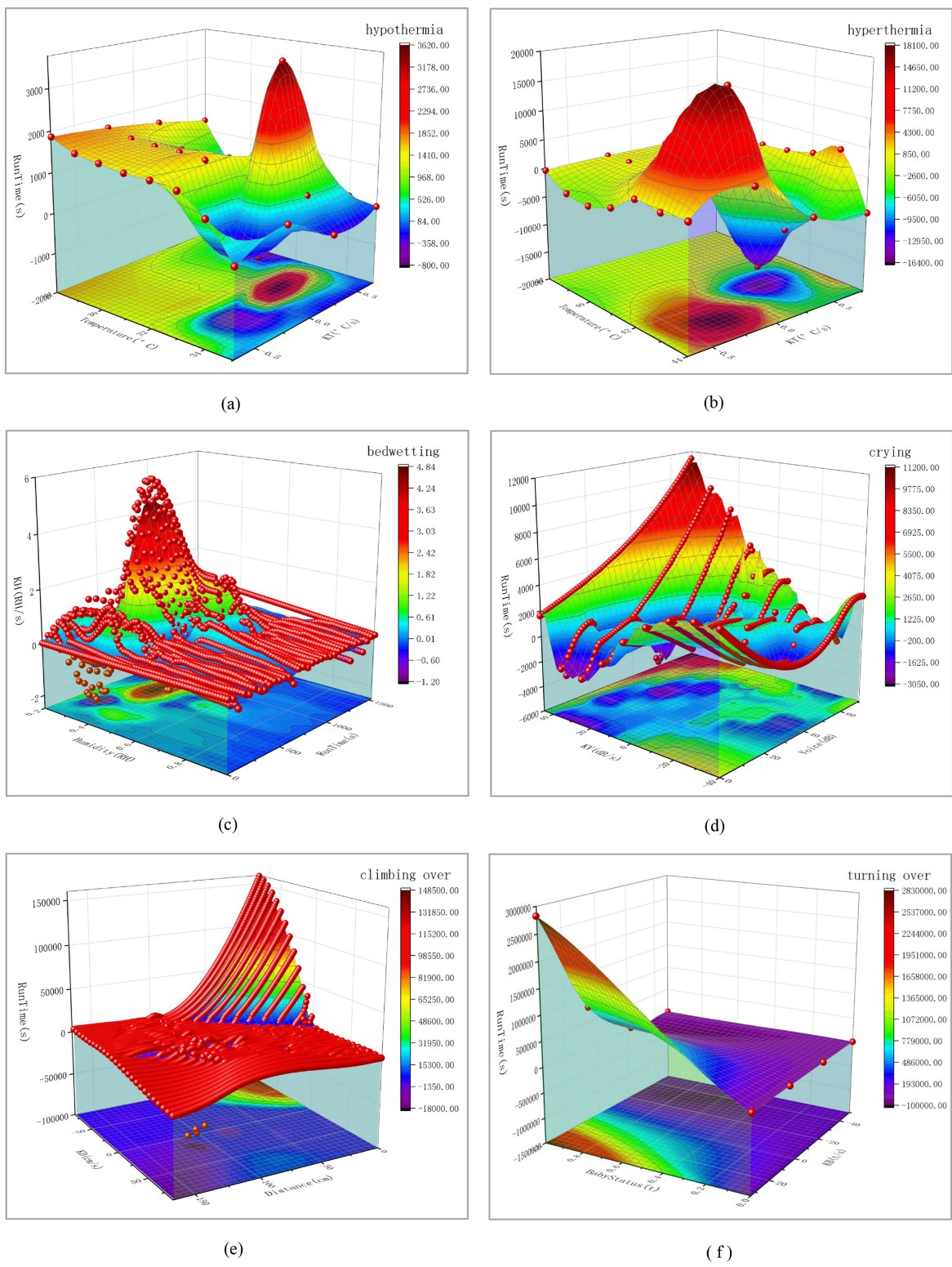

**Figure 4 Three-dimensional smooth data analysis chart for six hazardous scenarios.** Among them, the baby kicking open the quilt leads to hypothermia is (A), the infant fever leads to hyperthermia is (B), the infant bedwetting is (C), the infant crying is (D), the infant climbing over the crib leads to falling is (E), the infant turning over is (F).

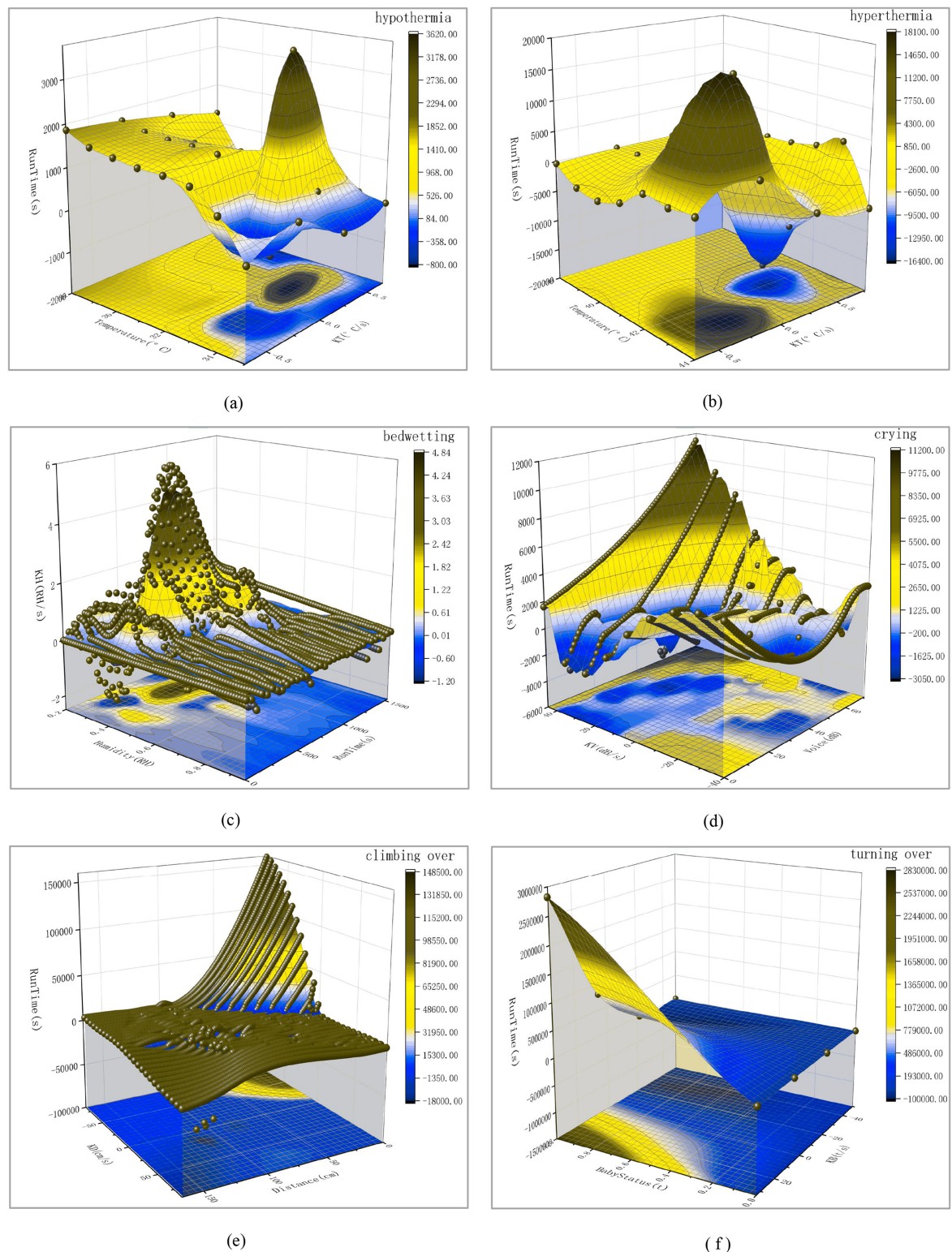

**Figure 5 Three-dimensional smooth data analysis chart for six hazardous scenarios.** (Red color blindness) Among them, the baby kicking open the quilt leads to hypothermia is (A), the infant fever leads to hyperthermia is (B), the infant bedwetting is (C), the infant crying is (D), the infant climbing over the crib leads to falling is (E), the infant turning over is (F).

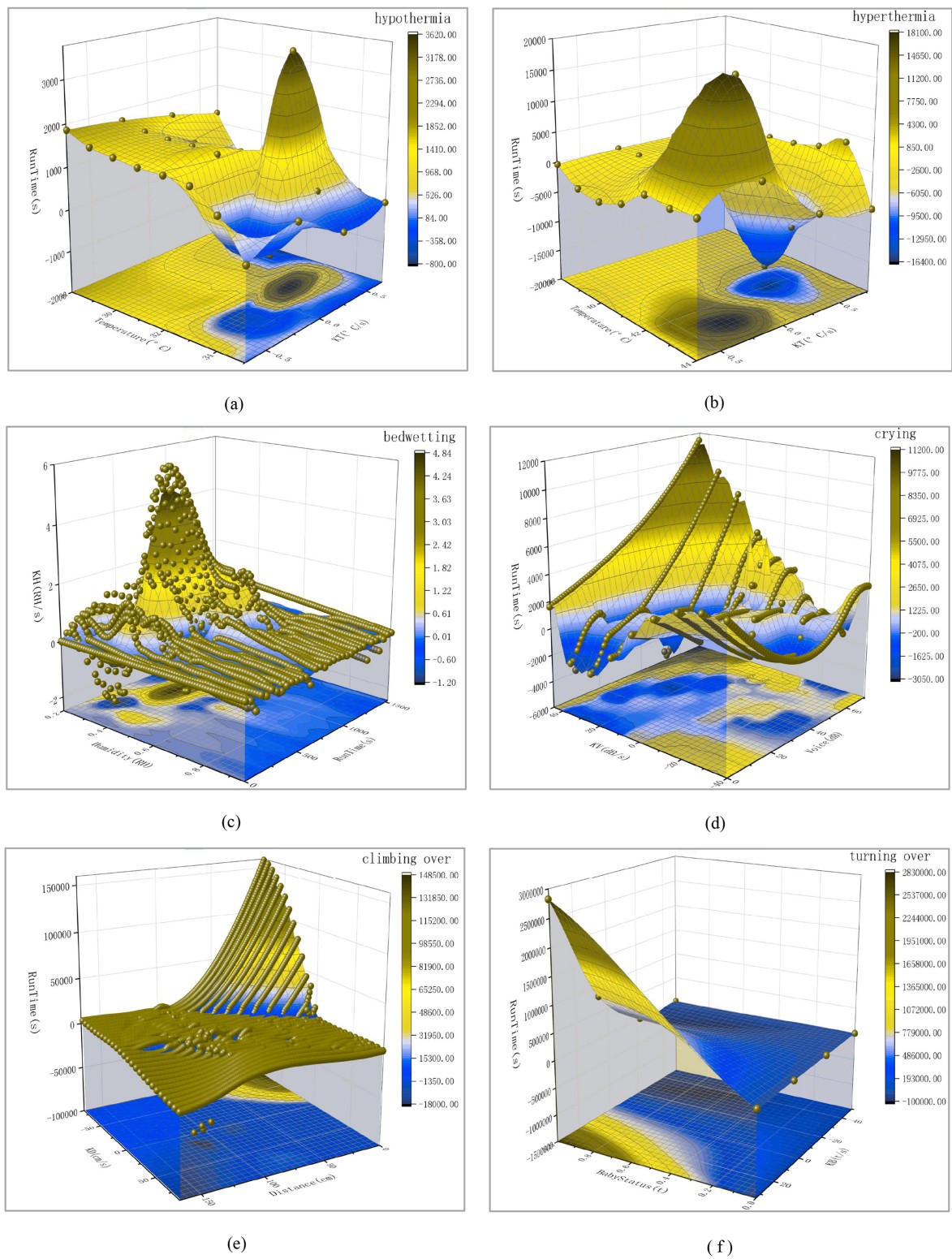

**Figure 6 Three-dimensional smooth data analysis chart for six hazardous scenarios.** (Green color blindness) Among them, the baby kicking open the quilt leads to hypothermia is (A), the infant fever leads to hyperthermia is (B), the infant bedwetting is (C), the infant crying is (D), the infant climbing over the crib leads to falling is (E), the infant turning over is (F).

**Table 2** This table contains values for various indicators of infants, such as run time (s), voice intensity (dB), distance (cm), temperature (°C), humidity (RH), and buzzer status.

| Run time (*s*) | Voice (*dB*) | Distance (*cm*) | Temperature (°C) | Humidity (*RH*) | Buzzer status |
|---|---|---|---|---|---|
| 82.84 | 46 | 9 | 37 | 31% | Non |
| 84.12 | 46 | 9 | 37 | 30% | Non |
| 85.4 | 46 | 0 | 37 | 25% | Ring |
| 86.9 | 46 | 0 | 37 | 24% | Ring |
| 88.4 | 46 | 51 | 37 | 27% | Non |
| 89.69 | 46 | 45 | 37 | 24% | Non |
| 90.97 | 45 | 120 | 37 | 24% | Non |
| 92.25 | 45 | 122 | 37 | 25% | Non |
| 93.54 | 45 | 121 | 38 | 28% | Non |

data, only two appeared, with an anomaly rate of 0.1807%, which has a small impact and can be ignored.

### *Test the detection function of the infant kicking open the quilt*

In this test, the system will be able to detect whether the baby has kicked open the quilt and issue a warning signal through a buzzer. The most common definition of hypothermia is body temperature ≤36.0 °C (*Ramgopal et al., 2023*). During the testing process, the temperature difference between the inside and outside of the quilt will be used to distinguish.

When the baby kicks off the quilt, the temperature sensor is exposed and the temperature begins to decrease. When the temperature drops below 36.0 °C (<36.0 °C), the buzzer will sound to inform parents to cover the quilt. The test results indicate that the system can issue timely warning signals when a baby kicks off the quilt. The function test of simulating a baby kicking off the quilt was successful, as shown in Fig. 1E.

Under the premise of ensuring that other factors fluctuate within a small range (not exceeding the set threshold), 1,126 pieces of hyperthermia data were collected for preprocessing, and the calculation is based on the following Eqs. (2) and (3):

$$T = (U/Vs) + b, \tag{2}$$

among them, $T$ is the temperature (°C), $U$ is the sensor output voltage ($V$), $V_s$ is the sensor sensitivity (constant), and $b$ is the sensor zero offset (constant).

$$KT = \Delta T/\Delta t = (T2 - T1)/(t2 - t1), \tag{3}$$

among them, $KT$ is the rate of temperature change, usually represented by the symbol $\triangle T/\triangle t$, in units of (°C/s), where $\triangle T$ is the amount of change in object temperature in degrees Celsius (°C), $\triangle t$ is the amount of change in time, which can be in seconds (s).

By performing formula calculations on 1,126 pieces of hypothermia data, three sets of valid information can be obtained: Temperature (°C), KT (°C/s), and Run Time (s).

Next, input these three sets of data into $A(X)$, $B(Y)$, and $C(Z)$ in the book table of Origin software, convert the worksheet into an $XYZ$ grid matrix, and then use the 3D Smoother

tool in Apps to smooth the data. The Smoother method selects the Adjacent Average, outputs the 3D image, and then projects the image to obtain the bottom image. Add the side semi-transparent fill color, adjust the scale line, input the meaning of the coordinate axis, name the image (upper right corner), and export it to PNG format. Finally, the 3D analysis graph is obtained, as shown in the following Fig. 4A.

From the above three-dimensional smooth image, it is not difficult to see that when the baby's body temperature is too low, the Temperature (°C) changes between 28 and 35 with the passage of Run Time (s), and the temperature change rate KT (°C/s) only changes between −0.75 and 0.75. This can effectively prove that the hazard prediction system can timely issue warning signals when monitoring the baby kicking off the quilt.

### Test the detection function of the infant fever

The purpose of this test is to check if the system can monitor infant fever. The temperature value can be displayed in real-time through the LCD1602A screen. According to the widely used definition of fever (temperature ≥38.0 °C) (*Kasbekar et al., 2021*), the system will trigger a buzzer alarm when the temperature inside the quilt rises to above 38.0 °C, reminding parents that the baby may have a fever and need timely treatment. During the testing process, to maintain a constant humidity, only increasing the temperature and simulating a baby's fever can be done using the hot air mode of a hair dryer or a cup with a lid filled with hot water for simulation. The test results show that the system can accurately monitor the temperature of infants and give timely alarms. The simulation test of infant fever function is considered successful, as shown in Fig. 1F. In this experiment, 1,283 hyperthermia data were collected for preprocessing, using the same formula as Eqs. (2) and (3).

By performing formula calculations on 1,283 hyperthermia data points, three sets of valid information can be obtained: Temperature (°C), KT (°C/s), and Run Time (s). Next, using the method described in the first test, a three-dimensional analysis diagram was finally obtained, as shown in Fig. 4B. From the above three-dimensional smooth image, it is not difficult to see that when the baby's body temperature is too high, with the passage of Run Time (s), the Temperature (°C) changes between 38 and 44, and the Temperature Change Rate KT (°C/s) only changes between −0.75 and 0.76. This can effectively prove that when the hazard prediction system detects a baby's fever, the temperature change is more significant, and it can issue a timely warning signal.

### Test the detection function of the infant bedwetting

In this test, the system will be checked to whether it can detect a baby wetting the bed and issue a safety warning signal through a buzzer. During the testing process, the towel will be moistened with warm water at around 37 °C, and the temperature will be controlled within the normal range of 36 °C to 38 °C. Only the humidity will be changed to simulate the baby wetting the bed. When using a wet tissue to wrap the humidity sensor, it is evident that the humidity value on the LCD1602A display screen rapidly soars to over 90%. The test results show that when the system detects a baby wetting the bed, the reading of the humidity sensor will exceed the threshold of 90%, and the buzzer will sound to notify parents to

replace the diaper promptly. The function test of simulating the baby bedwetting was successful, as shown in Fig. 1G.

In this experiment, 1,162 bedwetting data were collected for preprocessing, and the formulas used for calculation are as follows Eqs. (4) and (5):

$$H = (U - C3)/(C1 + C2 * R), \tag{4}$$

among them, $H$ is the relative humidity (RH); $U$ is the output voltage of the sensor (V); $C_1$ to $C_3$ are constants; $R$ is the output resistance of the sensor ($\Omega$).

$$KH = \Delta H/\Delta t = (H2 - H1)/(t2 - t1), \tag{5}$$

among them, $KH$ is the rate of change in humidity, usually represented by the symbol $\Delta H/\Delta t$, in units of (RH/s), where $\Delta H$ is the amount of change in relative humidity, in units of relative humidity (RH), and $\Delta t$ is the amount of change over time, which can be expressed in seconds (s). By performing formula calculations on 1,162 bedwetting data, three sets of valid information can be obtained: Humidity (RH), KH (RH/s), and Run Time (s). Next, using the method described in the first test, a three-dimensional analysis diagram was finally obtained, as shown in Fig. 4C.

From the three-dimensional smooth image, it is not difficult to see that as the Run Time (s) increases, the humidity (RH) changes between 20% and 95%, while the humidity change rate KH (RH/s) only changes between −2 and 6. This effectively proves that the hazard prediction system monitors the rapid changes in humidity when the baby wets the bed and can issue timely warning signals.

### Test the detection function of the infant crying

In this test, the system will be checked to see if it can detect a baby crying and issue a safety warning signal through a buzzer. During the testing process, an audio clip of a baby crying will be played on the phone to simulate a baby crying. When a baby wakes up and starts crying, the surrounding environment changes from quiet to noisy. The sound sensor receives the sound signal of a baby crying. When the reading exceeds the threshold of 50 dB, the buzzer will sound to notify parents to comfort their children. The test results show that the system can accurately monitor the crying situation of infants, with a monitoring accuracy rate of over 95%. The functional test of simulating baby crying was successful, as shown in Fig. 1H.

In this experiment, 1,303 crying data were collected for preprocessing, and the formulas used for calculation are shown in Eqs. (6) and (7):

$$V = 20 * log(U/U0), \tag{6}$$

among them, $V$ is the sound intensity (dB), $U$ is the sensor output voltage (V), and $U_0$ is the reference voltage (V).

$$KV = \Delta V/\Delta t = (V2 - V1)/(t2 - t1), \tag{7}$$

among them, $KV$ is the rate of change in sound intensity, usually represented by the symbol $\Delta V/\Delta t$, expressed in dB/s, where $\Delta V$ is the amount of change in sound intensity, expressed in decibels (dB), and $\triangle t$ is the amount of change over time, which can be

expressed in seconds (*s*). By performing formula calculations on 1,303 crying data, three sets of effective information can be obtained: Voice (*dB*), KV (*dB/s*), and Run Time (*s*). Next, using the method described in the first test, a three-dimensional analysis diagram was finally obtained, as shown in Fig. 4D.

From the three-dimensional smooth image, it is not difficult to see that as the Run Time (*s*) increases, the Voice (*dB*) changes between 0 and 70, and the sound intensity change rate *KT* (*°C/s*) only changes between −40 and 45. This effectively proves that the hazard prediction system can detect significant changes in sound intensity when a baby is crying, and can issue timely warning signals.

### *Test the detection function of preventing the infant from climbing over crib*

In this test, it will be checked whether the system can prevent babies from climbing over the crib. When there is no object around the crib, the distance measured by the ultrasonic sensor is a constant value. When the baby climbs near the edge of the crib, the distance displayed by the ultrasonic sensor on the monitor quickly shortens. When the distance value is less than 5 *cm*, it is determined that the baby is at the edge of the bed, and the buzzer will sound, notifying parents to handle the dangerous situation promptly. The test results show that the intelligent baby crib can effectively prevent babies from climbing over the crib, and the protective effect is very good. During the experiment, the function test to simulate preventing infants from climbing over the crib was successful, as shown in Fig. 1I. In this experiment, 1,255 climbing over data were collected for preprocessing. The calculation was based on the Eqs. (8) and (9):

$$D = K/(U - C), \tag{8}$$

among them, *D* is the distance (*cm*), *U* is the sensor output voltage (*V*), and *K* and *C* are the constants of the sensor.

$$KD = \Delta D/\Delta t = (D2 - D1)/(t2 - t1), \tag{9}$$

among them, *KD* is the rate of change in distance, usually represented by the symbol $\Delta D/\Delta t$, in units of (*cm/s*), where $\Delta D$ is the amount of change in distance in centimeters and $\Delta t$ is the amount of change in time, which can be in seconds. By performing formula calculations on 1,255 climbing over data, three sets of effective information can be obtained: Distance (*cm*), *KD* (*cm/s*), and Run Time (*s*). Next, using the method described in the first test, a three-dimensional analysis diagram was finally obtained, as shown in Fig. 4E. From the 3D smooth image above, it is not difficult to see that as the baby approaches the edge of the bed, the distance (*cm*) changes between 0 and 175 with the passage of Run Time (*s*) and the Distance Change Rate *KD* (*cm/s*) changes between −80 and 85. This effectively proves that the hazard prediction system can issue timely warning signals when the baby approaches the edge of the bed and the distance value is less than 5 *cm*.

### Test the detection function of the infant turn-over

In this test, the system will be checked to see if it can detect a baby turning over. When the turn-over detection algorithm cannot capture the baby's face within 5 s, it is judged that

**Table 3 The comparison table of change rates.**

| Hypothermia | Hyperthermia | Bedwetting | Crying | Climbing-over | Turning-over |
|---|---|---|---|---|---|
| −75~75% | −75~76% | −2~6 | −40~45 | −80~85 | −50~32 |

the baby has turned over. To prevent suffocation, a buzzer will sound at this time, reminding parents to handle it promptly. The test results show that the system can accurately monitor the situation of infants turning over. The functional test of simulating a baby turn-over was successful, as shown in Fig. 1J.

In this experiment, 10,000 turning over data were collected for preprocessing. The calculation was based on the above Eqs. (1) and (10) below:

$$KB = \Delta B/\Delta t = (B2 - B1)/(t2 - t1), \tag{10}$$

among them, $KB$ is the rate of change in turn-over, usually represented by the symbol $\Delta B/\Delta t$, in units of ($t/s$), where $\Delta B$ is the amount of change in turn-over times, in units of ($t$), and $\Delta t$ is the amount of change in time, which can be in units of seconds ($s$). By calculating the formula for 10,000 turning over data, three sets of valid information can be obtained: Baby Status ($t$), $KB$ ($t/s$), Run Time ($s$). Next, using the method described in the first test, a three-dimensional analysis diagram was finally obtained, as shown in Fig. 4F. From the three-dimensional smooth image, it is not difficult to see that as the Run Time ($s$) increases, the Baby Status ($t$) of the baby changes between 0.0 and 1.0, and the rate of change in the turn-over state $KB$ ($t/s$) changes between −50 and 32. This effectively proves that the hazard prediction system can issue timely warning signals when the baby cannot detect a face when turning over.

## System evaluation

Analysis can be conducted based on the change rate data obtained from the six specific scenarios mentioned above to conduct a clear evaluation of the entire hazard prediction system. The comparison table of change rates is shown in Table 3. The following is a detailed evaluation of some state:

(1) Accuracy assessment:

① Wide range of data changes: Most variables in this system, especially "crying", "climbing-over" and "turning-over", have a large fluctuation range, which may lead to increased uncertainty and affect the overall prediction accuracy.

② Deviation risk: If the predicted value tends towards the limit of the range of variation, it may lead to a higher false positive or false negative rate, thereby affecting accuracy.

(2) Sensitivity assessment:

① Sensitivity may be high: for example, the rate of change associated with "climbing over" is extremely wide, indicating that the system may be sensitive to these state changes.

② Confusion effect: In some cases, such as when the rate of change of "hypothermia", "hyperthermia" and "bedwetting" is relatively small, it may affect the sensitivity of the system to these specific behavioral changes.

(3) Actual performance

① Hypothesis testing: If the system is applied in clinical testing and its prediction results are reviewed through historical data, the performance of the system in prediction can be known, and real cases can be used to calculate accuracy and sensitivity.

② Feedback loop: In practical applications, the system should have the ability to self-learn and adapt, continuously improving accuracy and sensitivity through feedback.

Overall evaluation: Overall, the accuracy and sensitivity of this hazard prediction system have been validated through simulation experiments. The experimental results show that the system has high sensitivity and accuracy in different scenarios, with a fast response speed. It can accurately predict the occurrence of danger in advance and remind researchers to handle it promptly. However, during the experiment, there were still some significant data fluctuations that affected the overall accuracy, which should be further considered for processing and optimization. In addition, the system also needs to be validated through real data, analyzing the performance of each variable and its actual impact on the prediction target. In practical applications, it is necessary to further optimize the system by combining clinical experience and big data analysis to ensure high predictive ability under different conditions.

## CONCLUSION

This article designs an intelligent multimodal fusion risk prediction system based on AIoT technology, mainly composed of IoT hardware devices and artificial intelligence multimodal and multidimensional hazard prediction algorithms. Taking a baby crib as an example, this system enhances the functionality of a traditional baby crib and transforms it into an intelligent baby crib. The system can detect the baby's condition in real time and predict upcoming dangers, such as kicking the quilt, bedwetting, fever, crying, crawling out of the crib, and turning over. Before danger occurs, the system can promptly issue warnings to parents, prompting them to address potential issues.

The experimental results show that the intelligent multimodal fusion danger prediction system has accurate and effective detection and warning performance in six dangerous scenarios and real-time monitoring. The system can predict danger in advance and promptly remind parents to handle it. The original hypothesis holds. Based on the research of previous researchers, this study combines AIoT technology with multimodal data to solve key problems in short-term infant care, providing real-time monitoring and proactive warning capabilities, successfully filling the gap in the current system, and providing effective methods and ideas for the development of more advanced hazard prediction systems in the future.

However, due to not yet finding volunteer families or medical institutions, the solution proposed in this article is still in the simulation testing stage, and its feasibility in real-world scenarios needs further research and verification. In addition, in this experiment, the application of artificial intelligence technology is limited to image processing. In the future, artificial intelligence technology will also be considered for training and learning other multidimensional data, truly realizing a multidimensional danger prediction system. In the research process, there are also some ethical and moral issues, especially the challenges that

may be faced in technological implementation and practical life verification. At the same time, the ethical impacts of research work, such as data privacy and security, also deserve further investigation.

In the future, with the development of artificial intelligence, the application prospects of AIoT hazard prediction systems will be even broader. This design will further optimize the selection and use of sensors and Turn-over Detection Algorithms to improve the stability and accuracy of the system. At the same time, the intelligence level of the system will be strengthened, such as using speech recognition technology to interact with infants or improving the system's predictive ability and adaptability through machine learning technology. In addition, as research deepens, future studies may collaborate with medical institutions to apply the system to the early diagnosis and treatment of infant diseases. By applying the danger prediction system to different fields of real life, continuously solving problems, optimizing the prediction system, and improving the system's robustness and adaptability. The system will also explore more intelligent application scenarios, such as emotion recognition (*Kim et al., 2020*), state analysis, baby behavior prediction, and even achieving "baby language translation". By combining multimodal and multidimensional information and utilizing large-scale model training for analysis, parents can understand the language of infants and improve their parenting experience.

### Funding
The authors received no funding for this work.

### Competing Interests
The authors declare that they have no competing interests.

### Author Contributions
- Jibin Yin conceived and designed the experiments, performed the experiments, analyzed the data, authored or reviewed drafts of the article, and approved the final draft.
- Jia'nan Zhao conceived and designed the experiments, performed the experiments, analyzed the data, performed the computation work, prepared figures and/or tables, and approved the final draft.
- Xiangliang Zhang conceived and designed the experiments, analyzed the data, authored or reviewed drafts of the article, and approved the final draft.

### Data Availability
The data and code files are available in the Supplemental Files.

### Supplemental Information
Supplemental information for this article can be found online at http://dx.doi.org/10.7717/peerj-cs.2404#supplemental-information.

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
