# Peer review of "Design of a hazard prediction system with intelligent multimodal fusion based on artificial intelligence & internet of things technology: taking a crib as an example"

_PeerJ Computer Science, doi:10.7717/peerj-cs.2404_

## Round 0.1 · original submission · Minor Revisions

Please address the comments of the reviewers

Reviewer 1 ·

Basic reporting

The paper displays a good understanding of the topic and the authors have made a good effort to present the information in a clear and concise manner
However, there are several areas where improvements are necessary for clarity, accuracy, and overall quality. Below are my findings:

Specific comments:
Line 46-47: Do you have a reference for this statement? It would be helpful to provide a citation to support this claim

Line 60: "Farré". There is a small formatting error in the reference. Please correct it
Line 61: Do you have a reference for this statement? It would be helpful to provide a citation to support this claim
Line 79: "... in abandoned childcare stations." It is not clear what you mean by this. Please clarify. Is it a station childcare station for abandoned children or a childcare station that has been abandoned? Obviously, the latter is not possible. Please rephrase this sentence for clarity
Line 80-81: "... has positive significance for the harmony of families ..." Is this a personal opinion or did you measure the significance of this? Please clarify
Line 111-116: What exactly does this review have to do with your topic? As far as I could see this paper only covers the topic of Smart Agriculture and does not make any references or statements to infant care
Line 151: Duplicate "®" symbol. Please remove one of them
Line 212: "AONI high-definition camera". Is this a specific brand or model? Please clarify
Line 353-354: It is not relevant to mention the function calls of a given library in the paper
Line 374: "haarcascade frontalface alt2.xml". Are these 3 files or one single file? Also you can remove the concrete file name. It is enough to name the Classifier
Line 468-472: This paragraph contains the same information but in 4 different sentences. Please combine them into one sentence
Line 649: "'baby language translation (translating baby language)'". The sentence in the parentheses is redundant. Please remove it
Algorithm 2 (Turn-over detection) Algorithm: The shown algorithm is neither readable nor helpful to understand the Turn-over Detection Algorithm. It is also called "TODO". Please provide a more detailed explanation of the algorithm
Figure 1: Some of the text in the figure is not readable. Please increase the font size or use a different color for better readability

General comments:
1. The paper is well-structured and well-written. However, there are some minor grammatical errors that need to be corrected. Please proofread the paper carefully to correct these error
2. The figures and code snippets are helpful in illustrating the concepts discussed in the paper. However, you should put them in the actual paper instead of the appendix. This will make it easier for the reader to understand the concepts you are discussin
3. Your figures and tables should be better formatted. For example the figure and its label should be on the same page and the label is usually below the figure. Also, the tables should have a caption above the

Experimental design

The paper lacks a clear hypothesis or research question.
It would be helpful to clearly state the research question or hypothesis at the beginning of the paper. That makes it easier for the reader to understand the purpose of the study and the results that are presented.
In the Related Work section you presented some good references and previous work that was done in the field. However, you did not clearly state how your work is different from the previous work. It would be helpful to clearly state how your work is different from the previous work and what new contributions you are making to the field
The paper lacks a discussion section. It would be helpful to include a discussion section where you can discuss the implications of your findings and how they relate to previous research. In said discussion section, you should also discuss the limitations of your study and the ethical implications of your work.
Also most of the implemented algorithms are threshold-based. Thus one could argue that the system is not very robust and adaptable to different scenarios. It would be helpful to include a discussion on the limitations of the threshold-based algorithms and how they could be improved in future work.
The system also does not seem to learn from previous data. Is the "AI" part then just the image processing part? It would be helpful to clarify this in the paper

Validity of the findings

The paper presents some interesting and innovative ideas for using AI in infant care.
However, the paper lacks a clear evaluation of the proposed system.
It would be helpful to include a detailed evaluation of the system to show how well it performs in practice.
This could include a comparison with existing systems or a user study to evaluate the system's effectiveness and usability.
The paper also lacks a discussion of the limitations and ethical implications of the proposed system.
Once a research question is clearly stated, it would make is easier to evaluate the findings and discuss the implications and validity of the results

Reviewer 2 ·

Basic reporting

1. Objectives clearly mentioned and implemented.
2. The Developed system demonstrate various scenarios
3. Algorithms implemented identified real time detection of movements and predict potential dangers using alarm system.
4. References properly mentioned all are relevant
5.Figures neat and understandable
6. Shared video clearly understand the idea and objectives(Raw data)
7. Result analysis graphs explained with explanation. data received from IOT Devices analysed by using appropriate algorithms.

Experimental design

1. Fulfilled Aims and Scope of the journal.
2. Developed research mainly focused on social problem and aware parents for taking baby care using this solution.
3. Experiment focused on real-time status of the baby:-
baby hypothermia, 1126 pieces of hyperthermia data were collected for preprocessing
baby hyperthermia :- By performing formula calculations on 1283 hyperthermia data points, three sets of valid information can be obtained Temperature, KT, and Run Time. sufficient data
baby bedwetting :- 1162 bedwetting data were used.
baby crying :- 1303 crying data were collected for preprocessing and calculated sound intensity (dB)
baby climbing over the crib:- 1255 climbing over data were collected for preprocessing and calculate Distance.
baby turning over:- 10000 turning over data were collected for preprocessing and rate of change in turn over calculated.

Validity of the findings

1. Benefit to literature is clearly stated.
2. Data sets used robust, statistically sound and results calculated controlled.
3. the system can detect the real-time status of the baby and predict
upcoming dangers, Before danger occurs, parents can be promptly reminded to handle it.
4. results shown is indeed accurate and effective in detecting and warning the six dangerous scenarios and real-time monitoring scenarios mentioned above.
5. for real life scenarios not mentioned results.

Additional comments

1. try to implement for real time scenarios and mention the comparative results.
2. Experiment is appreciated and useful.
3. cant mention cost and how to provide services/use.

Reviewer 3 ·

Basic reporting

It employs clear and professional English throughout, providing sufficient background and context in the introduction. The article structure is professional, with well-defined sections for introduction, related work, methodology, experiment, and conclusion. The results are self-contained and relevant to the stated hypotheses. The literature references appear to be adequate, supporting the background and context of the research. The paper includes appropriate figures and tables to illustrate key concepts and results. While formal definitions of some key terms and detailed proofs are not extensively presented in the main manuscript, the overall technical depth appears sufficient for the scope of the paper. The methodology is well-explained, and the experimental design is clearly laid out. Overall, the manuscript successfully meets the requirements for basic reporting, providing a comprehensive and well-structured presentation of the research on the AIoT-based hazard prediction system for infant care. CopyRetry

Experimental design

The manuscript presents original primary research that aligns well with the aims and scope of a computer science journal, focusing on the development of an AIoT-based hazard prediction system for infant care. The research question is well-defined, relevant, and meaningful, addressing the important issue of infant safety and parental assistance. The authors clearly state how their research fills an identified knowledge gap in the field of smart infant care systems.

The investigation appears to be rigorous and performed to a high technical standard. The methodology section outlines a comprehensive approach to system design, including hardware components and software algorithms. The experimental setup covers seven different scenarios, demonstrating a thorough testing process.

The methods are described with sufficient detail to allow replication. The authors provide pseudocode for key algorithms, explain the sensor configurations, and detail the data collection and analysis processes. The use of professional tools like Origin for data visualization adds to the technical rigor of the study.

Ethical considerations seem to be implicitly addressed through the focus on infant safety, though a more explicit discussion of ethical implications could strengthen this aspect.

Overall, the experimental design meets the criteria for this section, presenting a well-structured, replicable study with clear relevance to the field. The research demonstrates a high technical standard in both system development and experimental validation.

Validity of the findings

Based on my review of the manuscript, here's my evaluation for Section 3: Validity of Findings:

The manuscript focuses on presenting a replicable study with a clear rationale and potential benefit to the literature in the field of AIoT-based infant care systems.

The underlying data presented in the manuscript appear to be robust and statistically sound. The authors provide detailed experimental results for seven different scenarios, including temperature, humidity, sound intensity, and distance measurements. The use of 3D smooth images generated from the collected data adds to the validity of the findings. The control of variables during experiments is explicitly mentioned, indicating a controlled experimental environment.

The conclusions are well-stated and directly linked to the original research question of designing an intelligent multimodal fusion hazard prediction system for infant care. The authors limit their conclusions to supporting results, discussing the system's effectiveness in detecting and warning about various dangerous scenarios. They also acknowledge the limitations of their study, noting that the solution is still in the simulation testing stage and requires further verification in real-life scenarios.

The authors provide a clear path for future research, suggesting improvements in sensor selection, algorithm optimization, and potential applications in medical diagnosis, which demonstrates a thoughtful consideration of the study's implications and limitations.

Overall, the validity of the findings is well-established within the scope of the study, with conclusions appropriately linked to the research question and supported by the presented results.

---

## Round 0.2 · accepted · Accept

The article is corrected according to comments

Reviewer 1 ·

Basic reporting

no comment

Experimental design

no comment

Validity of the findings

no comment

Additional comments

The revisions significantly improve the clarity and depth of the paper. It is clear that you have put great effort into implementing the suggestions, and the overall quality of the work has greatly benefited as a result. Good job.